# Associations between Surface Deformation and Groundwater Storage in Different Landscape Areas of the Loess Plateau, China

**Zhiqiang Liu** [1], **Shengwei Zhang** [1,2,3,*], **Wenjie Fan** [4], **Lei Huang** [1,2], **Xiaojing Zhang** [1,2], **Meng Luo** [1], **Shuai Wang** [1] and **Lin Yang** [1]

1   College of Water Conservancy and Civil Engineering, Inner Mongolia Agricultural University, Hohhot 010018, China; 3254890400@emails.imau.edu.cn (Z.L.); huanglei@imau.edu.cn (L.H.); nmndzxj@imau.edu.cn (X.Z.); luomeng@emails.imau.edu.cn (M.L.); 2020202060018@emails.imau.edu.cn (S.W.); yanglin111@emails.imau.edu.cn (L.Y.)
2   Key Laboratory of Water Resources Protection and Utilization of Inner Mongolia Autonomous Region, Hohhot 010018, China
3   Key Laboratory of Large Data Research and Application of Agriculture and Animal Husbandry in Inner Mongolia Autonomous Region, Hohhot 010018, China
4   General Administration of Hydrology of Inner Mongolia Autonomous Region, Hohhot 010010, China; fanwenjie329@163.com
*   Correspondence: zsw@imau.edu.cn

**Abstract:** The Loess Plateau is an important grain-producing area and energy base in China and is an area featuring dramatic changes in both surface and underground processes. However, the associations between surface deformation and groundwater storage changes in different landscape types in the region are still unclear. Based on Sentinel-1 and GRACE (Gravity Recovery and Climate Experiment) data, this study monitored and verified the surface deformation and groundwater storage changes in different landscape types, such as those of the Kubuqi Desert, Hetao Irrigation District, Jinbei Mining Area, and Shendong Mining Area, in the Loess Plateau of China from 2020 to 2021. Through time series and cumulative analysis using the same spatial and temporal resolution, the associations between these two changes in different regions are discussed. The results show that: (1) the surface deformation rates in different landscape types differ significantly. The minimum surface deformation rate in the Kubuqi Desert is $-5\sim5$ mm/yr, while the surface deformation rates in the Hetao Irrigation District, the open-pit mine recovery area in the Jinbei Mining Area, and the Shendong Mining Area are $-60\sim25$ mm/yr, $-25\sim25$ mm/yr, and $-95.33\sim26$ mm/yr, respectively. (2) The regional groundwater reserves all showed a decreasing trend, with the Kubuqi Desert, Hetao Irrigation District, Jinbei Mining Area, and Shendong Mining Area declining by 359.42 mm, 103.30 mm, 45.60 mm, and 691.72 mm, respectively. (3) The surface elasticity deformation had the same trend as the temporal fluctuation in groundwater storage, and the diversion activity was the main reason why the temporal surface deformation in the Hetao Irrigation District lagged behind the change in groundwater storage by $1\sim2$ months. The measure of "underground water reservoirs in coal mines" slows down the rate of collapse of coal mine roof formations, resulting in the strongest time-series correlation between mild deformation of the surface of the Shendong mine and changes in the amount of groundwater reserves (R = 0.73). This study analyzes the associations between surface deformation and groundwater storage changes in different landscape areas of the Loess Plateau of China and provides new approaches to analyzing the dynamic associations between the two and the causes of changes in both variables.

**Keywords:** surface deformation; changes in groundwater reserves; Kubuqi Desert; Hetao Irrigation District; Jinbei Mining Area; Shendong Mining Area

## 1. Introduction

The Loess Plateau is one of the four chief plateaus in China and has the largest loess accumulation district in the world [1]. The Loess Plateau is rich in resources and provides

an important ecological habitat for biodiversity [2]. In recent years, the uncontrolled development of coal and groundwater mineral resources and population increases have resulted in the ecological environment deteriorating severely [3]. The ecological environment of the Loess Plateau is facing increasing challenges such as agricultural irrigation [4], coal mining [5], surface runoff reduction [5], and land desertification [6]. The Loess Plateau region has become a research topic in watershed hydrology and social resources [7]. Efficient and accurate detection of the associations between surface deformation and changes in groundwater storage is an essential part of sustainable development, including the protection of water resources and the safety of human settlements. Sustainable and high-quality development is China's core national strategy. The government has taken a series of measures to contribute to the sustainable development of regional resources [8,9]. These policies have made the irregular surface deformation in the Loess Plateau particularly significant, and the fluctuation in groundwater reserves has decreased [10,11].

The application of remote sensing science has improved the capacity of human beings to explore nature [12]. Compared with conventional field observation data, remote sensing science can assess large areas across multi-spatial and temporal scales and allows real-time observation of environmental conditions [13]. In the field of remote sensing geological ecology, radar remote sensing is widely used in measuring surface deformation [14]. It can monitor the historical processes of surface deformation and discover geological disasters in real-time. Interferometric synthetic aperture radar (InSAR) technology has been used to monitor the surface deformation of the Qinghai–Tibet Plateau and analyze the freeze–thaw cycle process of frozen soil [15]. The emergence of differential interferometric synthetic aperture radar (D-InSAR) technology has broadened the monitoring time range of InSAR [16]. The combination of multiple sets of D-InSAR forms SBAS-InSAR (small baseline subset interferometric synthetic aperture radar) technology, which improves observation accuracy of geological disasters such as mining earthquakes and mountain earthquakes [17]. In the field of remote sensing hydrological environment, GRACE gravity satellites are mostly used to monitor large-scale terrestrial water storage changes [18]. The change in terrestrial water storage is usually used to analyze the ecological water cycle and to monitor drought events [19]. Terrestrial water storage is composed of five parts: soil water, surface water, ice and snow water, groundwater, and canopy water. Soil water content directly affects the weathering of soil minerals and the leaching of vegetation nutrients from soil [20] and indirectly affects the change in canopy water storage of vegetation communities [21]. Seasonal ice and snow water have made an important contribution to alleviating regional water shortages and restoring ecosystems [22]. Groundwater, as an important source of water for human life, industrial development, and coal mining, accounts for 22.4% of the Earth's human-available freshwater resources [23]. However, due to the large differences in the types of human activities in the Loess Plateau [24], a single remote sensing technology cannot accurately explore the associations between surface deformation and groundwater storage changes.

To determine the associations between surface deformation and groundwater reserves, InSAR monitoring of California's Central Valley surface deformation found that groundwater recharge led to seasonal surface deformation [25]. The combination of microgravity and InSAR technology proved that groundwater exploitation is the main cause of surface deformation in Al-Ain arid region (UAE) [26]. The development of shallow groundwater resources has led to serious surface deformation in the coastal area of Tianjin, where the excessive extraction of groundwater has caused the aquifer to undergo compacted inelastic deformation [27]. Groundwater level recovery is the main reason for the surface uplift of closed coal mines [28]. Although the associations between surface deformation and groundwater in these areas is significant, there are still some challenges in terms of analyzing spatial and temporal scales, for example, when attempting to improve the dimensional resolution of GRACE gravity satellite monitoring of terrestrial water. Methods such as least squares regression [29], machine learning [30], and multi-scale geographically weighted regression [31] have improved the dimensional resolution for terrestrial water storage.

However, downscaling in groundwater storage changes only reflects the characteristics of regional water resources, while there remain great limitations in analyzing the associations between surface deformation and groundwater in the Loess Plateau. The Loess Plateau can be divided into four regions according to the types of human activities: the Kubuqi Desert, Hetao Irrigation District, Jinbei Mining Area, and Shendong Mining Area. A large amount of the sand in the Kubuqi Desert comes from the ten tributaries of the Yellow River, and the natural wind erosion is a serious disaster [32]. The Hetao Irrigation District uses drainage to irrigate farmland to increase crop yield, and there is great demand for crop water [33]. Coal mining has resulted in significant spatial variability in vegetation growth status in the Jinbei Mining Area [34]. Shendong Mining Area is the world's largest underground coal mine production area, and coal mining volume increases year by year [35]. However, the types of human activities in the Loess Plateau are complex, and it is impossible to quantitatively analyze the associations between surface deformation and groundwater storage changes in each region using the same temporal and spatial scale. The lack of understanding of these associations is not conducive to the coordination and sustainable development of regional water resources.

The main purpose of this study is: (1) to monitor and verify regional surface deformation using Sentinel-1 data combined with SBAS-InSAR technology; (2) to monitor and verify the downscaled groundwater storage change using GRACE data and the weighted downscaling; (3) to analyze the associations between surface deformation and groundwater storage in different regions of the Loess Plateau using the same spatial and temporal scale.

## 2. Date and Methods

### 2.1. Study Area

The terrain of the Loess Plateau displays significant undulations, with an altitude varying between 47 and 3059 m. The continental monsoon climate has resulted in the coexistence of arid and semi-arid zones on the Loess Plateau. The summer is hot, and the winter is cold, with the temperature varying between $-20\,^\circ$C and $20\,^\circ$C. The lowest annual precipitation is 120 mm/yr, and the highest monthly evapotranspiration is 2800 mm/month. The types of human activities in the region can be split into four characteristic areas: the Kubuqi Desert, Hetao Irrigation District, coal base in Jinbei, and Shendong Mining Area. The Kubuqi Desert is located on the southern bank of the Yellow River irrigation area of 139,000 km$^2$ and is the seventh-largest desert in China [36]. The irrigation area of the Hetao Irrigation District is 6803.4 km$^2$, and the amount of water diverted from the Yellow River is about 5 billion m$^3$/yr. The Hetao Irrigation District is a crucial commodity grain and oil production base in China [37]. The precipitation in the Jinbei Mining Area is 486 mm/yr. The area has many open-pit mines as well as underground mines. Coal mining in the region has both a long duration and a large reclamation area [38]. The raw coal production of Shendong Coal Mine is about 291.83 million t/yr. The construction of a "coal mine underground reservoir" commenced in 2009 [39]. The annual water demand in Shendong Mining Area is about 70 million m$^3$; the main source of water in this area is groundwater [40].

### 2.2. Data

#### 2.2.1. Experimental Data

Sentinel-1 (https://scihub.copernicus.eu, accessed on 1 May 2022) SLC data were collected from 1 January 2020 to 31 January 2022 and included 4 parallel orbits and a total of 231 images. The surface deformation of each region was inverted, and accuracy adjustment was carried out by using the homonymous points of the coinciding regions. The elevation data were derived from the STRM 30 m resolution DEM terrain dataset (https://srtm.csi.cgiar.org/srtmdata/, accessed on 5 May 2022). POD (precise orbit data) was obtained from the official Sentinel-1 website (https://scihub.copernicus.eu/, accessed on 3 May 2022).

GRACE (http://www2.csr.utexas.edu/grace/RL06.mascons.html, accessed on 15 May 2022) data were collected from January 2020 to January 2021 CSR Mascon RL06 Level 3 data [41]. Mascon three-level data were calculated from RL06 two-level spherical harmonic coefficient product, including C20, C30 replacement, first-order term correction, and GIA correction preprocessing [42].

January 2020 to January 2021 monthly data were provided by the Noah submodule in the FLDAS [43] model, including ice and snow water data with an accuracy of 0.1°, four-layer soil water data with a depth of 0~2 m, and groundwater runoff data. Canopy water and surface water data were derived from January 2020 to January 2021, provided by the Noah sub-module in the GLDAS [44] model, with a spatial resolution of 0.25°. The IDW (inverse distance weighting) method was used to interpolate the original data to a spatial resolution of 0.01°.

The precipitation and evapotranspiration data were derived from the daily dataset of China's surface climate data (V3.0), which includes the annual, monthly, and daily average precipitation and evapotranspiration levels for each province, city, and region in China from 2002 to 2022 (20:00~20:00 cumulative precipitation and evapotranspiration, 0.1 mm unit). Raster data with a spatial resolution of 0.01° from January 2020 to December 2021 were generated using IDW spatial interpolation.

2.2.2. Measured Verification Data

Surface deformation data were verified using the measured level data of different periods from the Jinbei Mining Area and Shendong Mining Area, obtained at the mine scale, and the measured points are shown in Figure 1. In the Jinbei Mining Area, the four-level leveling method was used as deformation verification data to measure the relative deformation of the leveling points A1~A20 on 27 March 2020, 25 July 2020, 25 January 2021, and 21 May 2021, respectively. In the Shendong Mining Area, based on the known leveling points, fourth-order leveling was also used. The elevations of 4 December 2020 and 4 December 2021 leveling points were measured, respectively, and the relative deformation of each known point in 2020~2021 was obtained by subtracting the two elevations.

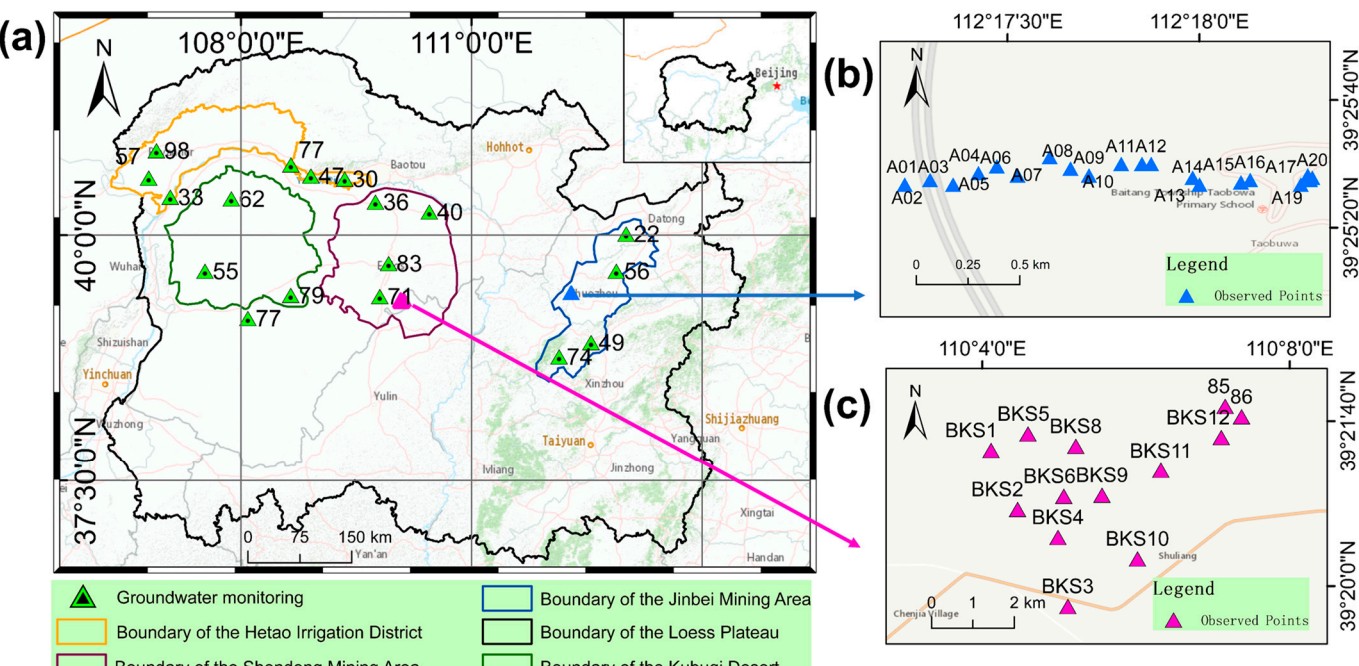

**Figure 1.** The geographical location of the study area. (**a**) Loess Plateau; (**b**) fourth-class leveling observed points in Jinbei Mining Area; (**c**) fourth-class leveling observed points in Shendong Mining Area.

The groundwater verification data from 2020 to 2021 were from a total of 37 groundwater well sites in the Inner Mongolia Water Conservancy Department. Preprocessing of the measured groundwater level data was carried out by deleting the missing well data (continuous missing time exceeded 12 months) and deleting outliers in well data where the measured groundwater level change was abnormal due to human factors. If large numbers of measured well data were adjacently distributed, and there were still some abnormally measured well data, the adjacent well data were also deleted. The spatial distribution in groundwater wells after preprocessing is shown in Figure 1.

*2.3. Method*

2.3.1. SBAS-InSA Technology

In 2002, Paolo Berardino first proposed SBAS-InSAR [45], and the inversion deformation accuracy was improved to the millimeter level [46]. In the $t_0 \sim t_n$ time series of the study area, $N$ scenes of SAR images are selected, and one scene is selected as the main image. The temporal and spatial baseline thresholds are set, and the remaining images are used for interference registration. The decoherence phase is eliminated, and the $M$-scene interferogram is obtained, which satisfies the following:

$$(N+1)/2 \leq M \leq N(N+1)/2 \tag{1}$$

The $j$th differential interferogram is generated by using the SAR image obtained by $t_a$ and $t_b$ $(t_a > t_b)$, and the interference phase can be expressed as the following:

$$\delta\varphi_j = \varphi_b(x,r) - \varphi_a(x,r) \approx [d(t_b,x,r) - d(t_a,x,r)]4\Pi/\lambda \tag{2}$$

where $1 \leq j \leq M$ represents the coherent phase; $\lambda$ represents the wavelength; and $d(t_a, x, r)$ and $d(t_b, x, r)$ represent the radar line-of-sight deformation at $t_a$ and $t_b$, respectively. According to the image acquisition time, the average surface deformation rate of the mining area can be obtained as follows:

$$V^T = \left[ V_1 = \varphi_1/(t_1 - t_0), \ldots, V_n = \varphi_{(n-1)}/\left(t_n - t_{(n-1)}\right) \right] \tag{3}$$

The differential phase is the following:

$$\sum_{(j=S_K+1)}^{(E_K)} \left(t_j - t_{(j-1)}\right) v_j = \delta\varphi_k, k = 1, 2, \ldots, m \tag{4}$$

where $E_K$ and $S_K$ are the main and slave image acquisition time, respectively, and $v_j$ is the pixel deformation rate at time $j$. This establishes the linear equation for the differential phase and the image time. The matrix equation is:

$$Av = \delta\varphi \tag{5}$$

Through the singular value decomposition method, the phase value of the deformation rate can be obtained, as well as the cumulative linear deformation of the surface.

A total of 231 images of four orbits and 11 image locations (Figure 2a) were selected in the Loess Plateau region, and SBAS-InSAR processing was performed using SARscape 5.6. The basic parameters were set as follows: the time baseline was 300 d, the spatial baseline was 300 m, and the coherence threshold was 0.2. Then, the interferogram was generated (Figure 2b,c, taking the fifth image position as an example). Taking the first scene image of each orbit in January 2020 as the super main image, 83, 85, 84, 86, 89, 87, 89, 84, 79, 85, and 87 interference pairs were generated, respectively. After eliminating the interferograms with poor coherence, a multi-view coefficient of 1:5 was set in the range and azimuth directions to improve the processing interference quality. POD (precision orbit data) were used to remove the orbit error, and 30 m SRTM DEM data were used to remove the terrain effect. The minimum cost flow method was used to unwrap the phase, and

the threshold of the unwrapped coherence coefficient was 0.3. The time series of surface deformation with an initial spatial resolution of 20 m was used for verification. Time cubic spline interpolation and spatial resampling were performed successively to make the temporal resolution of the surface deformation be by month and the spatial resolution 0.01° (≈1km).

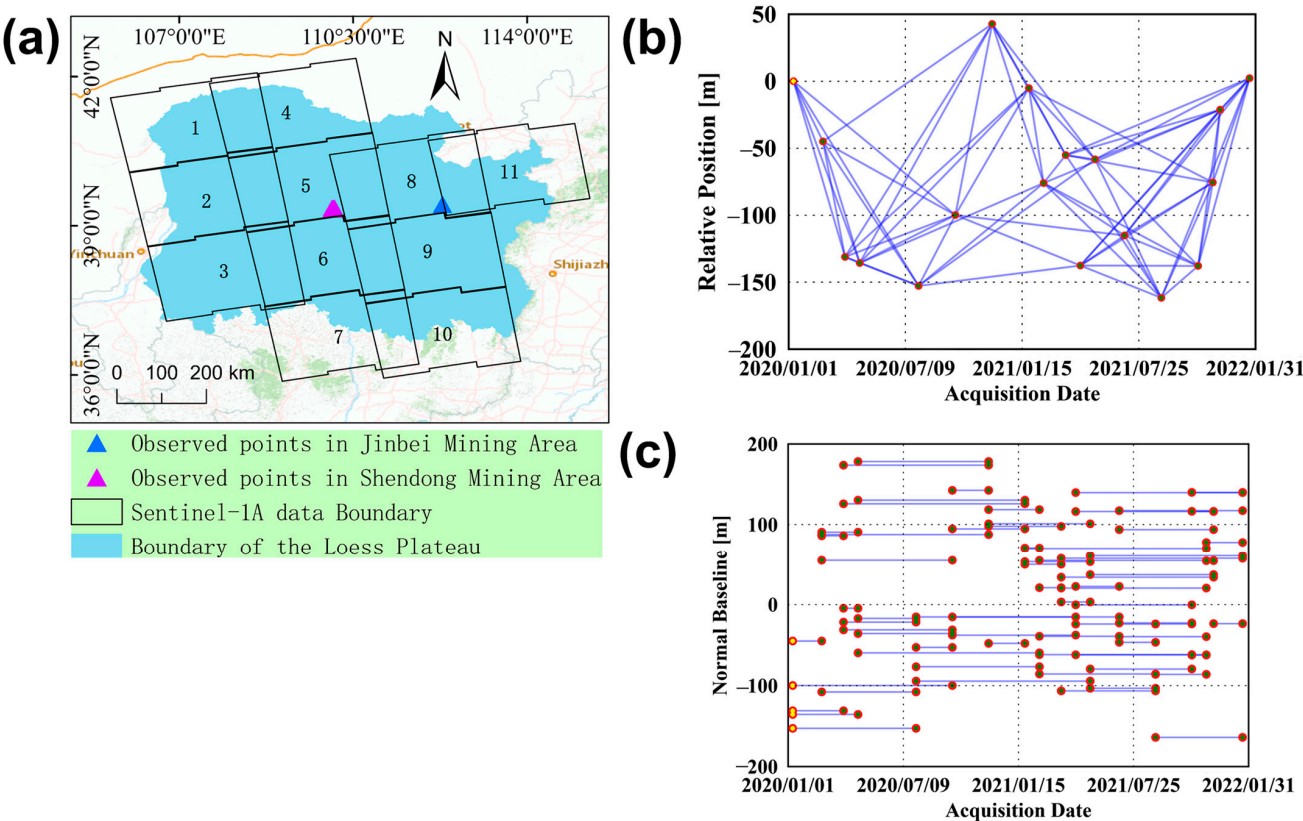

**Figure 2.** Sentinel-1 satellite image acquisition and spatio-temporal interference baseline. (**a**) Sentinel-1A data boundary and observed points; (**b**) Sentinel-1A 5th view image spatial interference pair information; (**c**) Sentinel-1A 5th view image temporal interference pair information.

### 2.3.2. Weighted Downscaling

The weighted downscaling method was first proposed by PENG Shouzhang et al. [47,48] and applied to the downscaling of precipitation products. Liu Hongliang et al. [49] compared the downscaling methods for GRACE data and found that the accuracy of the weighted downscaling method was significantly higher than that of multiplicative downscaling and error-allocation downscaling. The terrestrial water in the 0.01° model consisted of 0~2 m deep soil water, snow and ice water, groundwater runoff, canopy water, and surface water, with the data obtained from the FLDAS and the GLDAS, respectively (FLDAS represents 0.01° of the terrestrial water). In this study, downscaling was performed by the GRACE CSR RL06 Mascon three-level product with a spatial resolution of 0.25°. From 1 to 12 months in 2020–2021, 12 target grids were obtained by averaging by months.

$$TG_m^{025\_layer} = Mean\left( \sum_{y=2020}^{2021} TG_{y,m}^{025} \right) \tag{6}$$

$$TF_m^{001\_layer} = Mean\left( \sum_{y=2020}^{2021} TF_{y,m}^{001} \right) \tag{7}$$

In the formula, $TG_{y,m}^{025}$ represents the change in terrestrial water storage of the GRACE data in a specific year $y$ and month $m$, and $TG_m^{025\_layer}$ indicates that the resolution is 0.25°

according to the GRACE target grid. $TF_{y,m}^{001}$ represents the terrestrial water storage change in the FLDAS model, and $TF_m^{001\_layer}$ represents the specific target grid of the FLDAS at a spatial resolution of $0.01°$. Subtracting each month from the target grid of the corresponding month, one obtains $E_{y,m}^{025}$, as follows:

$$E_{y,m}^{025} = TG_{y,m}^{025} - TG_m^{025\_layer} \qquad (8)$$

$E_{y,m}^{025}$ is the target deviation for a specific month and year. The 12 target grids obtained by the hydrological model at the target resolution were combined with the interpolation results of the errors obtained by GRACE to obtain the downscaling results of terrestrial water storage.

$$TD_{y,m}^{001} = TF_{y,m}^{001\_layer} + f\left(E_{y,m}^{025}\right) \qquad (9)$$

where $TD_{y,m}^{001}$ represents TWSA downscaled to $0.01°$ spatial resolution; $f\left(E_{y,m}^{025}\right)$ the interpolation result produced $0.25°$ spatial resolution.

### 2.3.3. Changes in Groundwater Reserves

The change in terrestrial water storage after downscaling was examined using evapotranspiration and precipitation data, and the reasonableness of the $0.01°$ terrestrial water storage change was analyzed. The $0.01°$ groundwater storage change was calculated based on the water balance principle. Terrestrial water storage (TWS) is composed of five forms of water. They are soil water storage (SMS), surface water storage (SWS), canopy water storage (CWS), snow water storage (SWE), and groundwater storage (GWS). The composition of terrestrial water storage is:

$$TWS = SMS + SWS + SWE + CWS + GWS \qquad (10)$$

According to the water balance principle, (10) can be transformed into the following:

$$\triangle TWS = \triangle SMS + \triangle SWS + \triangle SWE + \triangle CWS + \triangle GWS \qquad (11)$$

In the formula, $\Delta$ indicates that the monthly variation of the current month is obtained by subtracting the previous month. The Loess Plateau region $\Delta GWS$ is as follows:

$$\triangle GWS = \triangle TWS - (\triangle SMS + \triangle SWS + \triangle SWE + \Delta CWS) \qquad (12)$$

### 2.3.4. Verification of Groundwater Storage Change

After multiplying the measured groundwater change $\Delta H$ after pretreatment with the regional aquifer-specific yield $S_n$, the measured groundwater level change in the region is obtained:

$$\Delta GWS = \Delta H S_n \qquad (13)$$

The specific yield $S_n$ is assumed by various factors such as natural precipitation, soil infiltration rate, and aquifer thickness, and differs across the regions of the Loess Plateau. The northern part of the Kubuqi Desert is adjacent to the Yellow River, with a feed rate of 0.06, and the southern part's feed rate is only 0.007 due to the uneven distribution of groundwater [50]. The specific yield of Kubuqi Desert is 0.015. The specific yield of the weak permeable layer in the Hetao Irrigation District is 0.02~0.03, the specific yield of the phreatic layer and the confined water layer is 0.025~0.04, and the specific yield along the Yellow River is 0.06 [51]. The measured groundwater in the Hetao Irrigation District includes the phreatic water layer and confined water layer, and the value of specific yield is 0.03. The Middle Ordovician karst groundwater is the main aquifer in the Jinbei Mining Area, and the specific yield of the aquifer is 0.024 [52]. The bedrock pore fissure aquifer in the Shendong Mining Area is the phreatic section of the Middle Jurassic Anding Formation and the phreatic section of the Zhiluo Formation. The water yield of the landscape aquifer is 0.022 [53].

2.3.5. Data Processing of Surface Deformation and Groundwater Storage Change

Combined with the research of Li C [54] and the actual field investigation and analysis, the surface deformation grade was divided into micro deformation, mild deformation, moderate deformation, and severe deformation. In the Hetao Irrigation District and the Kubuqi Desert on the Loess Plateau, 100 points were selected, and 100 points were selected in the micro-deformation (0~20 mm/yr), mild deformation (20~50 mm/yr), moderate deformation (50~100 mm/yr), and severe deformation (>100 mm/yr) categories in the Jinbei Mining Area and the Shendong Mining Area. After averaging points with the same conditions, surface deformation and groundwater storage changes were analyzed for temporal and cumulative changes at the same spatial and temporal scales (spatial resolution of 0.01°, temporal resolution of 1 month). Temporal change refers to the increase or decrease in the current month's change compared with the previous month, and cumulative change refers to the cumulative change from the base month to the target month.

## 3. Results and Analysis

### 3.1. Surface Deformation Verification and Feature Analysis

3.1.1. Surface Deformation Verification

The SBAS-InSAR results of 27 March 2020, 25 July 2020, 21 January 2021, and 21 May 2021 in the Jinbei Mining Area were compared with the leveling data. Using 27 March 2020 as the starting time, the relative morphometric variables of the corresponding image points were calculated using the nearest-image method (averaging the surface morphometric variables within a square with a side length of 200 m centered on the actual measurement point). The measured deformation and inversion results are shown in Figure 3a,c,e and Figure 3b,d,f as scatter plots. Point A1 to A20 are linearly arranged in line with the direction of the coal mine roadway excavation, and point A20 is close to the coal mining center. The maximum error of a single point was 12.9 mm, the minimum error was 1.5 mm, the median error was 5.4 mm, and the average error was 5.6 mm.

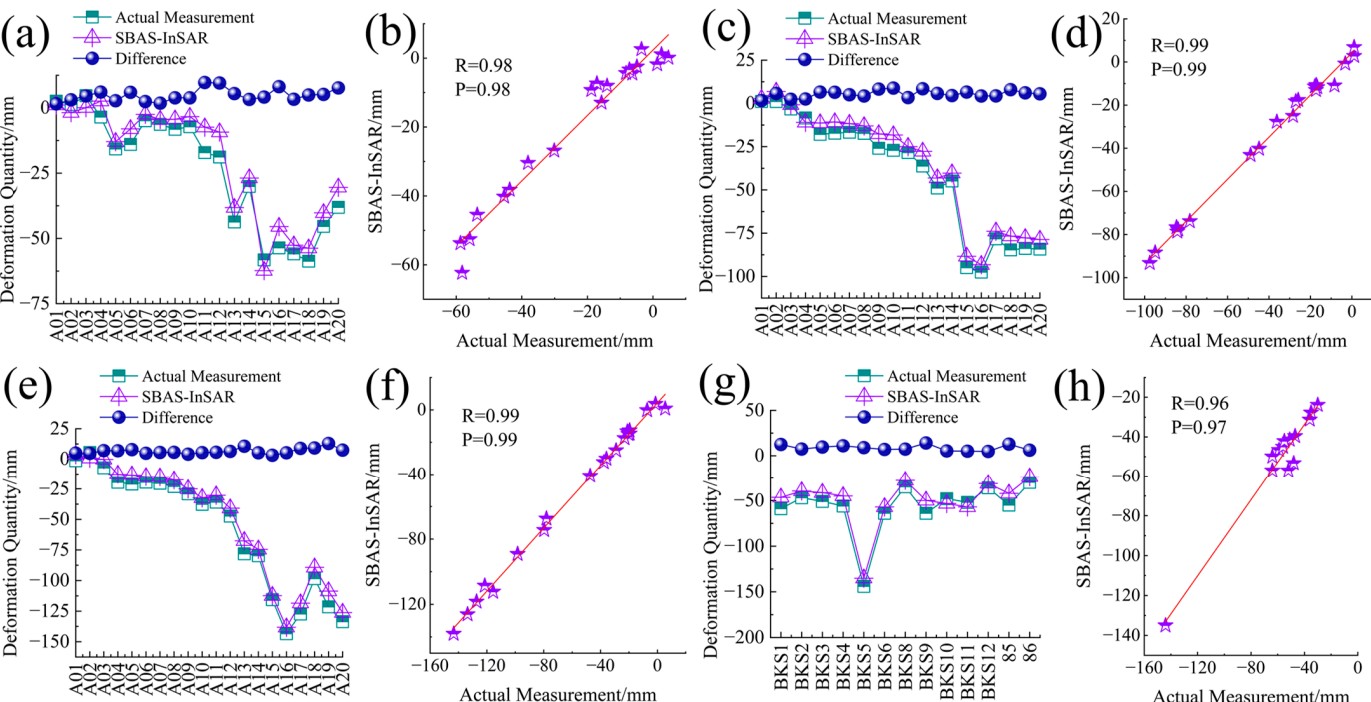

**Figure 3.** Deformation verification. (**a**) 25 July 2020 Jinbei Mining Area surface deformation; (**c**) 21 January 2021 Jinbei Mining Area surface deformation; (**e**) 21 May 2021 Jinbei Mining Area surface deformation; (**g**) 4 December 2021 Shendong Mining Area surface deformation; (**b**,**d**,**f**,**h**) is a scatter plot of (**a**,**c**,**e**,**g**).

The cumulative deformation maps of the Shendong Mining Area from 27 November 2020 and 4 December 2021 were selected. Based on the image on 27 November 2020, the cumulative deformation maps of the two scenes were subtracted to obtain the relative deformation from 2020 to 2021. Through the known point coordinates of the mining area, the two groups of deformation variables were compared according to the nearest pixel method (Figure 3g; the scatter plot is Figure 3h). The maximum single-point error was 14.08 mm, the minimum error was 5.3 mm, and the average error was 7.44 mm.

### 3.1.2. Analysis of Surface Deformation

The surface deformation of the Loess Plateau is strong throughout the region (Figure 4). A positive value indicates uplift, and the negative value indicates subsidence. The deformation rate of the Kubuqi Desert was −5~5 mm/yr. The deformation rate in the irrigation area was between −60 and 25 mm/yr, and the area with a deformation rate of −25~15 mm/yr accounts for more than 50% of the total area. Irrigation had a notable influence on surface deformation, and there was no geological disaster found in the irrigation area.

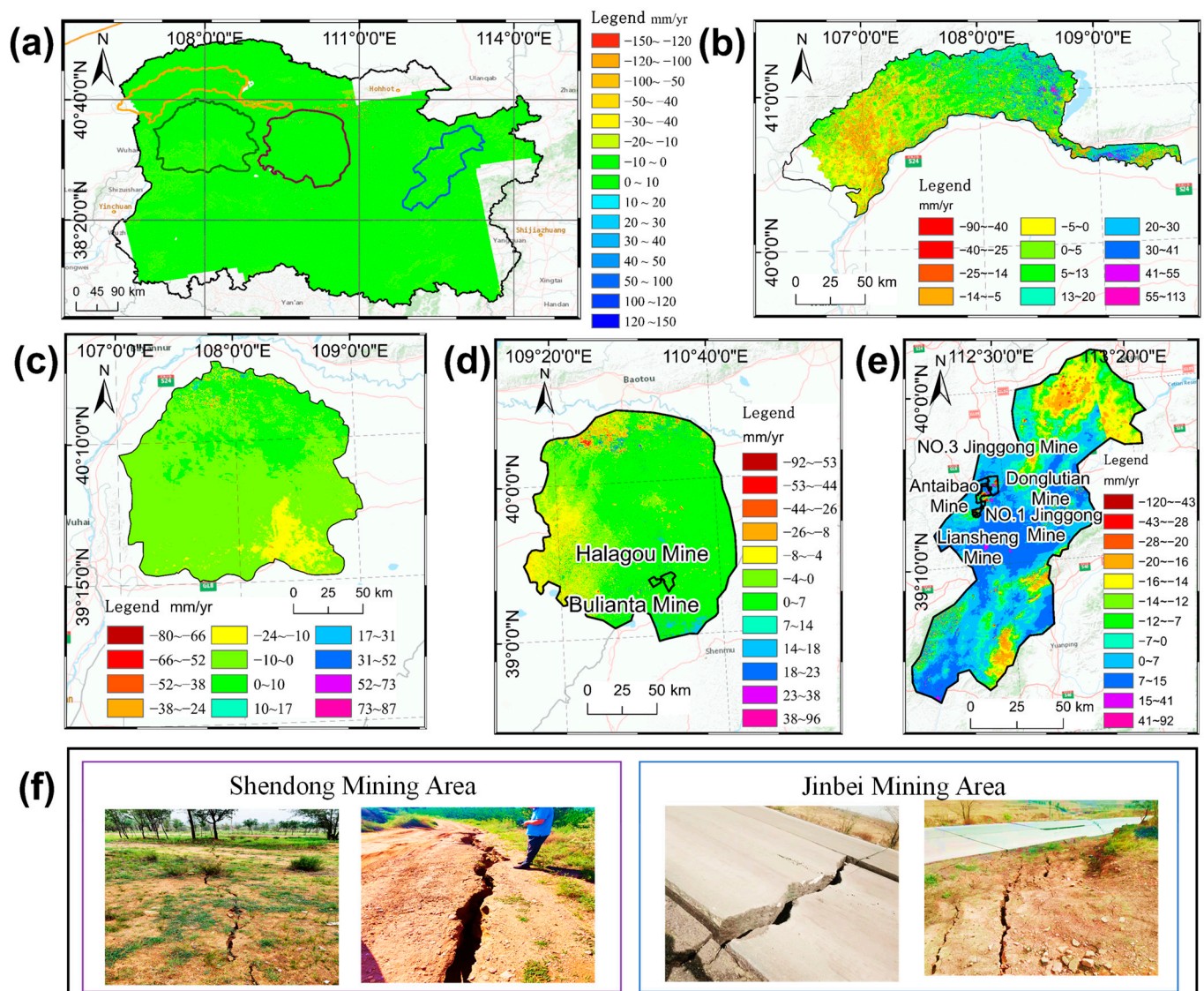

**Figure 4.** Annual variation in surface deformation. (**a**) Loess Plateau; (**b**) Hetao Irrigation District; (**c**) Kubuqi Desert; (**d**) Shendong Mining Area; (**e**) Jinbei Mining Area; (**f**) earth fissures and geologic hazards in the Jinbei and Shendong Mining Areas.

The annual surface deformation of Dangxin Mine and Jinggong No. 3 Mine in the Jinbei Coal Mine Base was between −45 mm and −30 mm. The maximum deformation at the junction of No. 1 Mine and Liansheng Mine was −339 mm, forming a settlement funnel centered at the junction. There were many ground fissures and land collapses, and geological disasters were prevalent (Figure 4f). The surface deformation rate of the mining stripping area in the East Open-Pit Mine was −30~26 mm/yr. The soil compaction in the mining recovery area of the Antaibao Open-Pit Mine caused surface deformation, and the maximum deformation was −158 mm. The average uplift of the stripping area, the pile area, and the dump was 25 mm/yr.

The maximum deformation of the Bulianta Mine in the Shendong Mining Area was −177 mm, and the surface subsidence area was 3.3 km². The minimum area of surface deformation in the Halagou Mining Area was 2.27 km². The surface subsidence funnel of underground coal mining was rectangular, and the surface deformation in the excavation direction of the coal mining roadway was significant. In the process of inclined shaft mining, the settlement funnel was bowl-shaped, and the surface deformation process usually fluctuated, displaying settlement following uplift (Figure 4f).

### 3.2. Validation and Analysis of Groundwater Storage Change

#### 3.2.1. Validation of Groundwater Storage Change

The measured groundwater level change data were quantified and compared with the downscale groundwater reserve change. Well data from the six measured points were selected in four regions (Figure 1). The variation in groundwater storage in a square with a side length of 10 km was quantified with the measured data as the center. The average value of groundwater storage change was processed and compared with the measured groundwater level change (Figure 5). From the analysis, it can be seen that the correlation coefficient R between the change in measured groundwater depth and the change in groundwater reserves is between 0.54 and 0.77 ($p < 0.01$). The change in groundwater reserves in the downscaled data displays the same trend as that in the measured groundwater depth, which proves that the results of the change in groundwater reserves in the downscaled data were reliable. The downscaled groundwater storage changes are relatively stable, whereas the measured groundwater depth changes greatly.

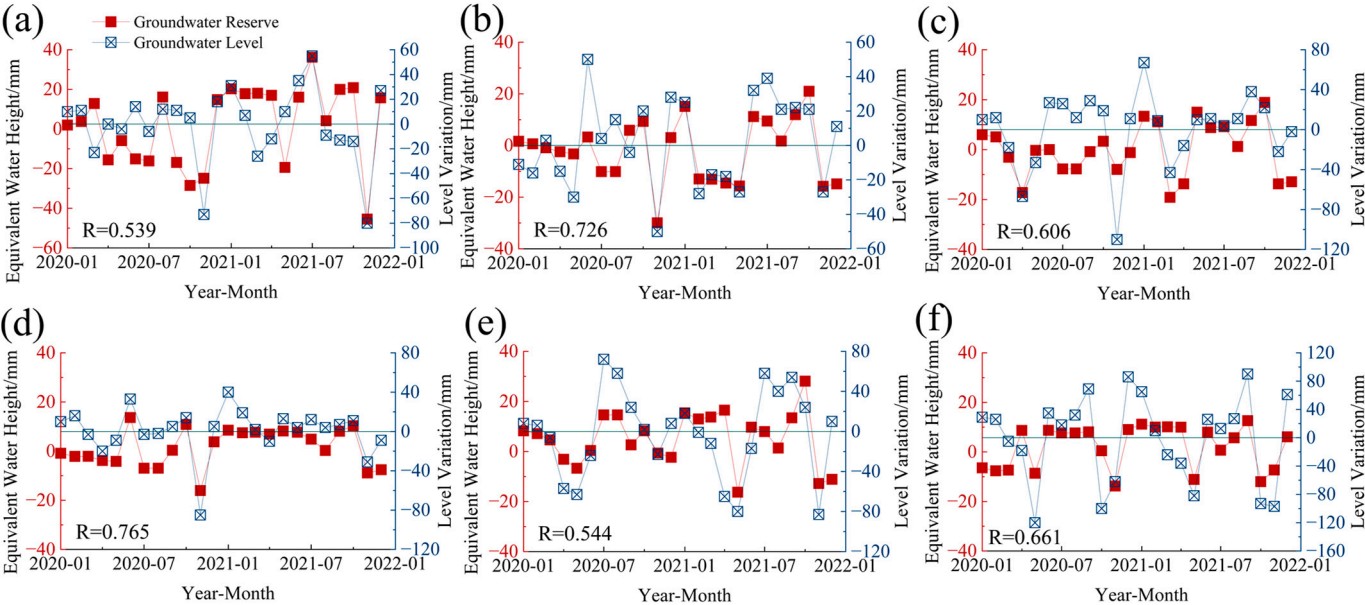

**Figure 5.** Verification of groundwater reserve; (**a**) is 36; (**b**) is 40; (**c**) is 49; (**d**) is 57; (**e**) is 62; (**f**) is 98.

### 3.2.2. Analysis of Groundwater Storage Variation

From 2020 to 2021, the total groundwater reserves in the Loess Plateau decreased by 20.914 billion m$^3$, and the changes in groundwater reserves in each region showed a decreasing trend (rate change, Figure 6; time change, Figure 7). The groundwater reserves in the Loess Plateau reduced by 234.17 mm in 2020 and 206.65 mm in 2021. The groundwater reserves in the Kubuqi Desert fluctuated between −46 and 5 mm. Groundwater reserves near the Yellow River in the northern part of the desert increased, with a rate of change of −1~5 mm/yr. The groundwater reserves in the southern part of the desert decreased significantly, with a rate of −43~−36 mm/yr.

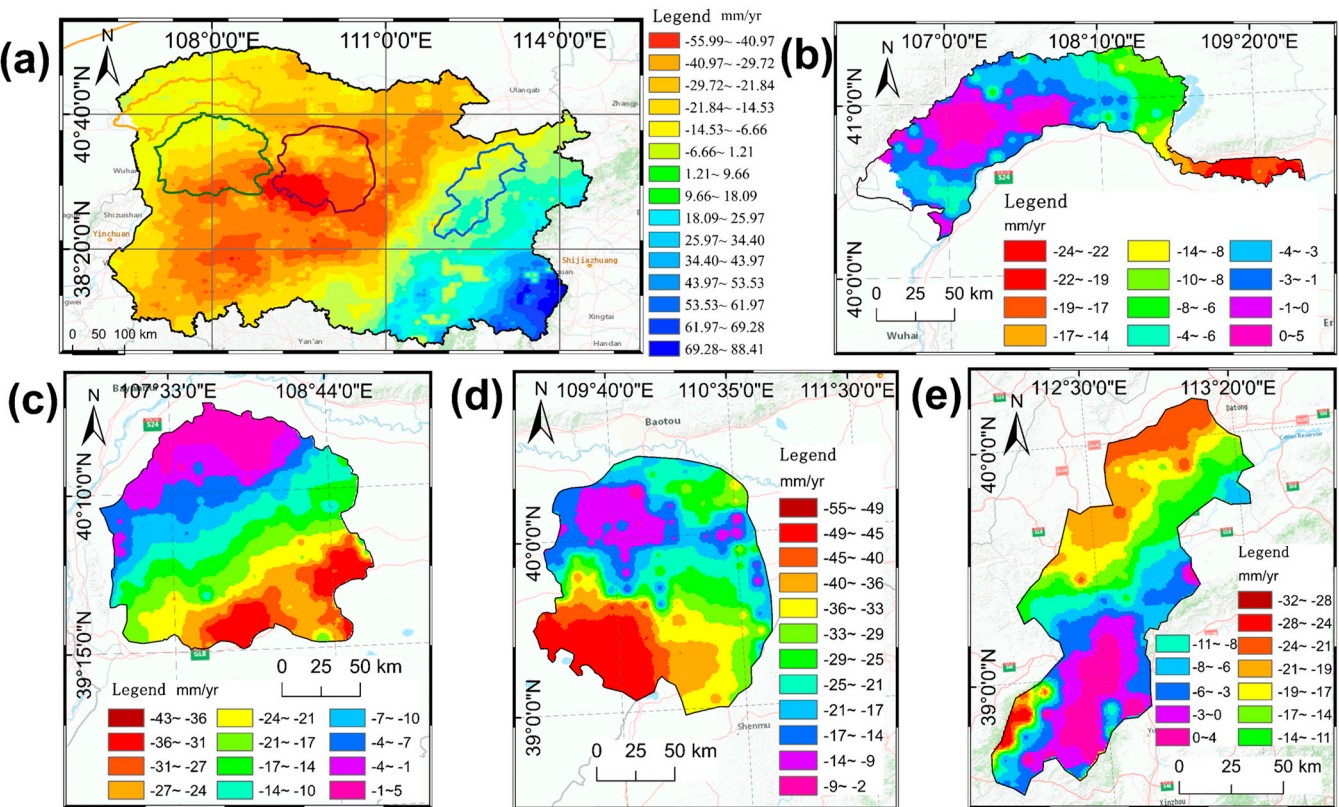

**Figure 6.** Annual variation in groundwater storage. (**a**) Loess Plateau; (**b**) Hetao Irrigation District; (**c**) Kubuqi Desert; (**d**) Shendong Mining Area; (**e**) Jinbei Mining Area.

The groundwater storage in the Hetao Irrigation District decreased by 5~24 mm/yr. The area where the groundwater storage fluctuates between −8 mm and 10 mm accounts for 80% of the Hetao Irrigation District. The groundwater reserves in the eastern part of the Hetao Irrigation District decreased significantly, with a cumulative decrease of 48~28 mm. Due to the influence of water diversion irrigation from April to September every year, the decreased trend in groundwater reserves in the Hetao Irrigation District from May to October was significantly diminished (Figure 7).

Groundwater storage in the Jinbei Mining Area varied in a range of −164~8 mm/yr. The groundwater storage in the northern part decreased by 34~164 mm/yr, and the groundwater storage in the southern part changed by −12~10 mm/yr. Due to the significant difference in the spatial distribution of precipitation, precipitation was mainly concentrated in the southern part of the Jinbei Mining Area, with a maximum precipitation of 316.7 mm.

The fluctuation in groundwater reserves in the Shendong Mining Area decreased significantly, with a range of −201.86~−190.56 mm. The groundwater reserves in the eastern part of Shendong Mining Area decreased by 150~170 mm. The groundwater storage in the southwest decreased significantly, with fluctuation of −156~−145 mm/yr.

The northwest region is close to the Yellow River, and the groundwater storage decreased slowly, with a change of −48~−24 mm.

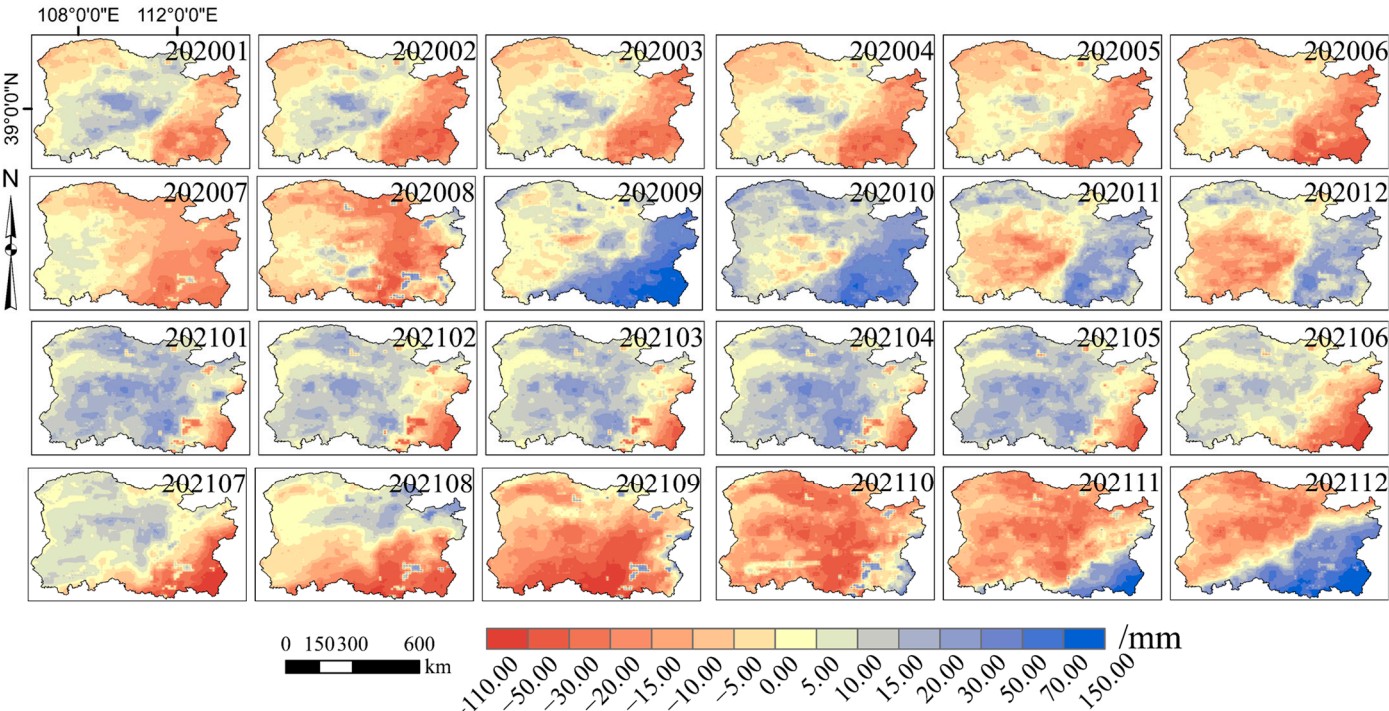

**Figure 7.** Downscaling spatial variation in groundwater reserve.

*3.3. Analysis of the Associations between Surface Deformation and Groundwater Storage Change*

The spatial difference between surface deformation and groundwater storage in the Loess Plateau is significant. The time series of groundwater reserves in the Kubuqi Desert in summer increased and fluctuated significantly, and the fluctuation in surface deformation time series was also significant (Figure 8A). From January to October 2020, the temporal variation in groundwater reserves fluctuated by 6.51 mm, the temporal variation in surface deformation was 2.88 mm, and the temporal variation showed high synchronization. According to comparative analysis of the cumulative changes (Figure 8a), the cumulative reduction in groundwater reserves in 2020 was significant, and the cumulative surface deformation fluctuated greatly (R = 0.51).

The time series of groundwater storage and surface deformation in the Hetao Irrigation District between May and October 2020 was pronounced (Figure 8B). The groundwater storage showed a decreasing trend from February to March 2021, and the surface exhibited significant subsidence from March to April. Cumulative deformation analysis showed that (Figure 8b) when the cumulative change in groundwater reserves fluctuated between September and December 2021, the cumulative surface deformation continued to decrease. There was a negative correlation between groundwater change and surface deformation when this cumulative change occurs, and the correlation coefficient was 0.49. The surface deformation of the irrigation area had a lag period of 1 to 2 months for the change in groundwater storage. The surface deformation lagged behind the change in groundwater reserves by 1~2 months. The lag had a correlation of R = 0.77 with a lag period of 1 month and a correlation of R = 0.42 with a lag period of 2 months.

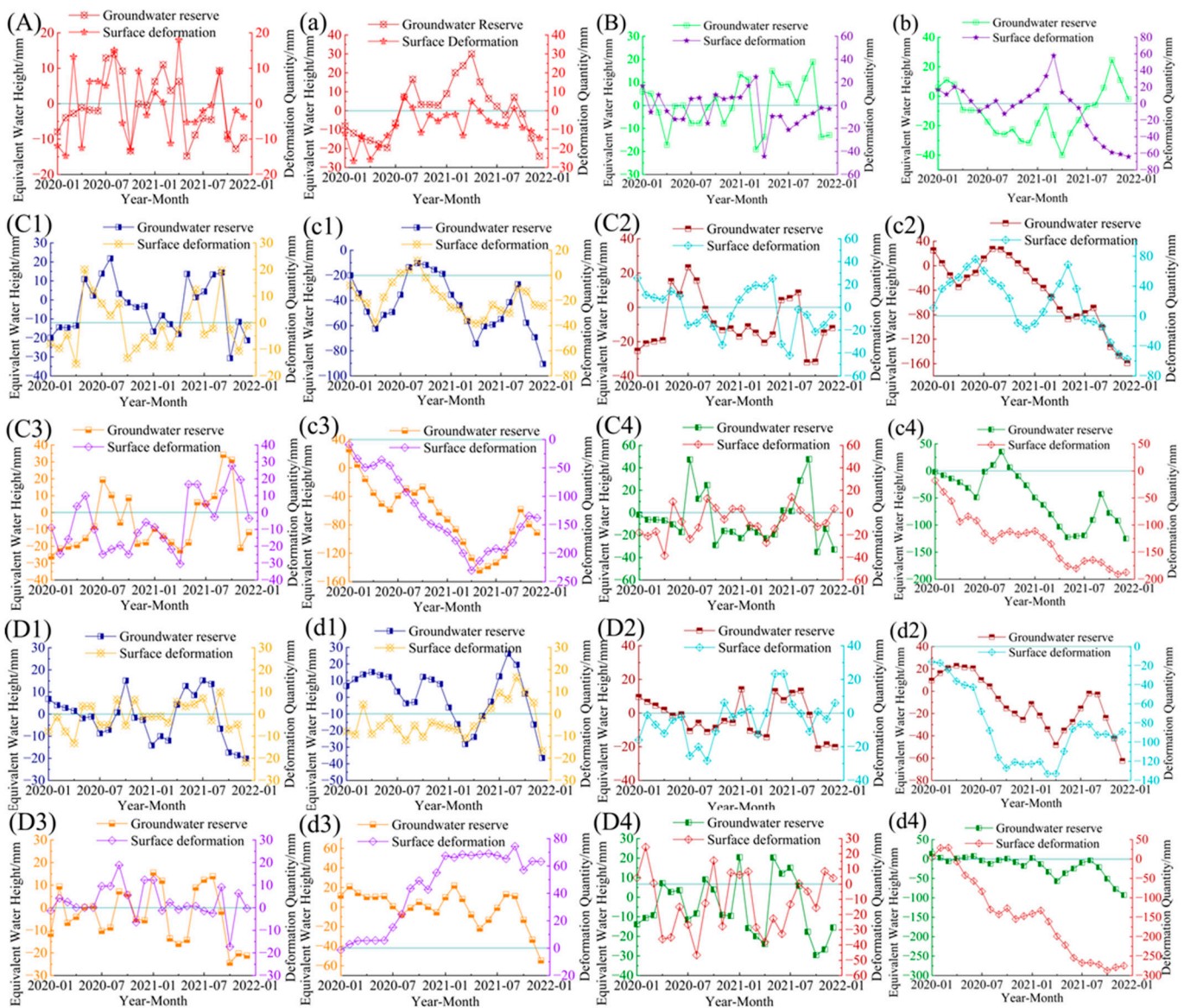

**Figure 8.** The associations between groundwater storage change and surface deformation. (**A**) Temporal changes in the Kubuqi Desert, (**a**) Cumulative changes in the Kubuqi Desert, (**B**) Temporal changes in the Hetao Irrigation District, (**b**) Cumulative change in the Hetao Irrigation District, (**C1**) Micro time series deformation in the Jinbei Mining Area, (**c1**) Micro cumulative deformation in the Jinbei Mining Area, (**C2**) Mild time series deformation in the Jinbei Mining Area, (**c2**) Mild cumulative deformation in the Jinbei Mining Area, (**C3**) Moderate time series deformation in the Jinbei Mining Area, (**c3**) Moderate cumulative deformation in the Jinbei Mining Area, (**C4**) Severe time series deformation in the Jinbei Mining Area, (**c4**) Severe cumulative deformation in the Jinbei Mining Area, (**D1**) Micro time series deformation in the Shendong Mining Area, (**d1**) Micro cumulative deformation in the Shendong Mining Area, (**D2**) Mild time series deformation in the Shendong Mining Area, (**d2**) Mild cumulative deformation in the Shendong Mining Area, (**D3**) Moderate time series deformation in the Shendong Mining Area, (**d3**) Moderate cumulative deformation in the Shendong Mining Area, (**D4**) Severe time series deformation in the Shendong Mining Area, (**d4**) Severe cumulative deformation in the Shendong Mining Area.

The micro-timeseries for deformation in the Jinbei Mining Area was positively correlated with the change in groundwater storage (R = 0.58). From September 2020 to April 2021, the cumulative groundwater storage decreased by 50.72 mm, and the cumulative settlement in surface deformation was 42.80 mm. When the mining area had more minor

deformation (Figure 8C1), the change in groundwater reserves was strongly correlated with topographic deformation, and there was a large consistency in variation between changes in groundwater reserves and surface deformation. Under mild deformation (Figure 8c2), the change in groundwater storage from August to December 2021 showed a decreasing trend, and the surface deformation fluctuated before settling down. The variation in cumulative change for both variables had a consistent and correlated trend, with R = 0.57. During the moderate time series deformation (Figure 8C3), the change in groundwater storage from July to October showed an increasing trend, and the difference between the two trends was notable. The groundwater storage for severe time series deformation (Figure 7C4) changed significantly with seasonality, and the positive correlation coefficient of the cumulative change was R = 0.72.

The micro-deformation of the surface deformation and the change in groundwater reserves in the Shendong Mining Area were significantly consistent during summer. The average fluctuation difference of the time series was less than 10 mm, and the correlation coefficient was 0.45 (Figure 8D1). Under mild time series deformation (Figure 8D2), the change in groundwater storage in 2020 showed a consistent decreasing trend, and the synchronous correlation between the two time series was 0.73. From April to July 2021, the slight cumulative deformation difference was large (Figure 8d2). When the groundwater storage increased by 32.97 mm, the cumulative change in surface deformation increased by 61.86 mm. From May to September 2021, the groundwater storage of moderate cumulative deformation showed an increasing trend, and the surface deformation only increased from August to September, with a correlation coefficient of R = −0.40. Both showed a decreasing trend in severe cumulative deformation (Figure 8d4), and the correlation coefficient R = 0.67. There was no significant fluctuation response between severe surface deformation and groundwater storage change.

In the associations between groundwater storage and surface deformation in the Loess Plateau, the Kubuqi Desert is the least disturbed by human activities, and the time series surface deformation is consistent with the fluctuation in groundwater storage. The Hetao Irrigation District has elastic deformation due to irrigation, and the geological structure has not changed. The surface deformation and groundwater storage change in the Hetao Irrigation District showed a time delay, and the cumulative fluctuation in the two is large. The fluctuation trend in groundwater reserves in the mining area was similar to that of slight deformation, but the decrease in groundwater reserves in the Shendong Mining Area was slow due to the influence of the "coal mine underground reservoir" [54].

## 4. Discussion

### 4.1. Associations between Surface Deformation and Groundwater Storage Change

The surface deformation of the Loess Plateau can be divided into elastic surface deformation and inelastic surface deformation [55]. The geological strata are subjected to in situ stress, and the geological structure state is unchanged. Surface deformation that returns to its original state after the stress disappears is called elastic surface deformation [56]. In situ stress changes the physical state of geologic rock strata, causing strata fracture and collapse, which is the dominant factor of inelastic deformation of the earth's surface and the direct cause of geologic hazards [57]. In the Kubuqi Desert, where anthropogenic disturbances are infrequent, fluctuations in groundwater water storage are the main cause of surface deformation in summer and fall (Figure 8A,a). The Kubuqi Desert has severe wind erosion hazards in winter and spring, and wind migration of sandy soils leads to elastic deformation of the surface [32]. Precipitation is the only input factor of the water cycle, and the groundwater storage in the Hetao Irrigation District increases after precipitation. The study [58] pointed out that the infiltration of precipitation caused the wetting effect, accelerating structural changes and the surface subsidence of the gully slope, and this seasonal groundwater change would cause seasonal surface elevation. The change in groundwater storage in the Hetao Irrigation District has no seasonal fluctuation due to the influence of irrigation factors (Figures 5 and 6), and the surface timeseries deformations

are consistent with the changing trend in groundwater (Figure 8B). The difference in surface deformation between drainage irrigation and pumping irrigation is significant. The pumping irrigation in North China [55], California [59], and other regions of the United States has led to the depletion of groundwater reserves and has caused geological disasters, whereas no geohazards occurred in diverted irrigation in the Hetao Irrigation District. Therefore, seasonal precipitation, wind erosion disaster, and water diversion irrigation are the main factors causing surface deformation to be elastic in the Kubuqi Desert and Hetao Irrigation District.

Mining of coal seams causes in situ stress redistribution in the airspace, and geostress alters the stable state of geologic rock formations [60]. Stress is transmitted to the ground, producing geological disasters such as ground cracks and ground collapse [30]. Coal mining in Jinbei is the main cause of surface deformation. The time series and cumulative associations between groundwater reserves and deformation grade were different (Figure 8C1–c4). The variation range and the trend in groundwater storage change on mild surface cumulative deformation and moderate surface time series deformations were consistent. The depletion of groundwater reserves in the Jinbei Mining Area was greater than that in the Shendong Mining Area. In 2009, the Shendong Mining Area constructed a coal mine underground water reservoir by transferring stored water, allowing for the protection of groundwater resources while coal mining [30]. The temporal variation in surface moderate deformation and groundwater reserves in the Shendong Mining Area was consistent with fluctuations (Figure 8D2,d2). The increase in coal mining volume and coal mining water consumption from February to June leads to the continuous decline in groundwater reserves [61]. The combined effect of coal mining activities and groundwater reserves changes produces surface deformation. Zheng Meinan et al. [28] noted that following the closure of the coal mine, the surface experienced uplift or subsidence deformation due to the combined action of groundwater and goaf collapse. There were significant uplifts in the heap soil area, the stripping area, and the mining recovery area of the Jinbei Mining Area (Figure 4). The stacking of mining soil and coal gangue in open-pit mines causes surface uplift [62]. During the operation of open-pit mines, the original supporting soil layer and rock are removed due to the forward spalling activity [63], resulting in an imbalance in surface stress changes and surface uplift. Most of the mining soil and rock backfill in the mining restoration area also produces surface uplift [64], while larger slopes and exposed rock surfaces will also experience surface uplift through rainfall and soil erosion [65].

### 4.2. Analysis of Influencing Factors in Surface Deformation and Groundwater Storage Change

The variation trend in groundwater reserves is related to natural recharge, climate-driven factors, spatial and temporal variation of runoff events, hydrogeological conditions, and the morphology of the region. Overexploitation and climate change lead to the depletion of groundwater due to the reduction in natural recharge [66]. The variation in groundwater storage in different regions of the Loess Plateau differs significantly (Figure 6). The change in groundwater storage in the irrigation area does not fluctuate seasonally according to irrigation factors. The groundwater in the Shendong Mining Area and Jinbei Mining Area is affected by mining factors, and the variation in groundwater reserves is reduced. Studies have shown that human intervention in the recharge–discharge system will accelerate the interaction between surface water and groundwater [67]. The water consumption of coal mining in the Shendong Mining Area and Jinbei Mining Area accounts for 94% and 97% of the impact of human activities [68]. The irrigation method in the Hetao Irrigation District comprises the introduction of water from the Yellow River water into the farmland; thus, the groundwater storage in the irrigation area always increased from September 2020 to September 2021. Rodell et al. confirmed that the groundwater recharge in an area of groundwater pumping irrigation decreased by about 50% [17,69,70]. Extracting groundwater irrigation will cause surface subsidence [54,71], and drainage irrigation (Figure 8b) causes surface fluctuation deformation; however, drainage irrigation

will not cause groundwater depletion that otherwise causes ground subsidence to form geological disasters. The time series of the Hetao Irrigation District changes from May to October 2020 (Figure 8B). Groundwater reserves have obvious fluctuations due to human irrigation factors and precipitation changes [72], and the trend in surface fluctuation deformation is the same as that in groundwater reserves.

Soil type and soil erosion degree differ wildly across the Loess Plateau. Different soil types have different infiltration rates. Infiltration is affected by precipitation, environmental conditions, and soil physical properties [73] and depends on soil permeability and water content [74]. One study [58] found that soil properties, including soil mechanical composition, soil density, soil initial water content, soil bulk density, and soil structure, are the main factors affecting precipitation infiltration. Studies have shown that precipitation above 35.00 mm can supplement soil water in the 60~80 cm soil layer [54]. The average precipitation input in July and August of 2021 in the Shendong Mining Area is only 33.5~41.7 mm, and the precipitation is extremely unstable, while the evapotranspiration is much higher than the precipitation. The serious shortage of precipitation is one of the important factors that led to the significant decrease in groundwater storage in the Shendong Mining Area from September to December 2021 (Figure 6). Coal mining produces ground fissures and water-conducting zones [75,76]. Water-conducting fissure zones accelerate the rate of water circulation, making the change in groundwater reserves vulnerable to seasonal changes (Figure 8D4,d4).

Surface deformation is closely related to changes in groundwater storage, and water reallocation causes surface deformation. By clarifying the correlation between the two, changes in groundwater reserves can be judged by surface deformation, which is conducive to regional water conservation. Regional development of water conservation and water use policies is more conducive to guaranteeing the sustainable use and management of water resources [77,78]. Diversionary diffusion irrigation is the main irrigation method in the Hetao Irrigation District [79], and drip and sprinkler irrigation methods can protect groundwater level stability [80]. Reducing elastic surface deformation also aids in ensuring the safety of the lives and property of the residents of the Hetao Irrigation District. The development of water-conducting fissure zones in mining areas accelerates the rate of groundwater storage reduction [81] and increases the risk of continued deterioration of geologic hazards [82]. The orderly management of water-conducting fissure zones and geologic hazards in mining areas is the key to minimizing the depletion of groundwater levels.

*4.3. Sources and Analysis of Errors in Surface Deformation and Groundwater Storage Changes*

Influenced by the orbit design of the Sentinel-1A satellite, the image revisiting period of the 40°N latitude region is 12~36 days [83], and only 1~3 views of images can be acquired between July~August in summer in the Loess Plateau. In this study, the time interval of the images was in the range of 24~36 days, and the lower image coverage reduces the image coherence and has an impact on the accuracy of surface deformation monitoring [84]. Due to the limitation of satellite orbit and lifetime, Sentinel-1B images could not be acquired in the Loess Plateau region to obtain three-dimensional surface deformation [85]. The radar line-of-sight direction is not exactly perpendicular to the surface, and even the nearest neighbor method still has some errors. The climatic environment of the Kubuqi Desert features large temperature differences and high wind speeds and is characterized by frequent turbulence and scattering phenomena, thereby increasing atmospheric errors [86]. Even when atmospheric error corrections are added or interferometric pairs with atmospheric errors are removed, tropospheric errors are still not completely negligible [87]. After temporal cubic spline interpolation to produce a temporal resolution of 1 month, studies have shown that time scale interpolation can have a degrading effect on the accuracy of the results [88]. Unifying the spatial resolution of surface deformation and groundwater storage changes to 0.01°, spatial resampling reduces the spatial resolution of surface deformation while increasing the generation of errors [89].

The variability of errors in different GRACE gravity satellite products varies between specific products. CSR RL06 Mascon data have the highest accuracy when obtaining changes in terrestrial water storage in the Loess Plateau region [53]. Mascon applies the spherical harmonic coefficient solution to reduce the stripe error and leakage error [90]. GRACE reflects the overall quality change in the region [31], and the overall quality change in the mining area includes the change in terrestrial water reserves and the change in coal mining quality. The change in coal mining quality is 1~2 orders of magnitude smaller than that in terrestrial water reserves, but this change, without deduction, may be a source of error. The correlation in groundwater storage changes in Shanxi and North China monitored by 0.25° GRACE is 0.70~0.79 [60,91]. The change of 0.01° groundwater storage in the Jinbei Mining Area and the measured groundwater level change is 0.726 (Figure 5c), which provides an advantageous higher spatial resolution. Groundwater storage changes include all aquifers in the region [92]. The correlation between the change in groundwater reserves and the measured groundwater level in the Kubuqi Desert and the Shendong Mining Area is 0.539~0.544 (Figure 5a,e), which is lower than that in the Hetao Irrigation District 0.661~0.765 (Figure 5d,f). This may be due to the incomplete monitoring of aquifers by regional groundwater level as well as uneven spatial distribution, ultimately resulting in errors [93].

## 5. Conclusions

The associations between surface deformation and groundwater storage in different landscape areas of the Loess Plateau differ significantly. The specific conclusions are as follows:

(1) There are significant regional differences in surface deformation landscape across the Loess Plateau. The surface deformation of the Kubuqi Desert with no change in geological rock properties is −10~10 mm. Under the influence of artificial irrigation factors, the surface deformation rate in the Hetao Irrigation District is −60~25 mm/yr. The surface deformation in the Kubuqi Desert and Hetao Irrigation District is elastic deformation. The deformation rate in the open-pit mine recovery area in the Jinbei Mining Area is −25~25 mm/yr, and the geological disasters in the first underground mine are more severe. The maximum deformation rate in the subsidence funnel in Shendong Mining Area is −95.33 mm/yr. The frequency of geological disasters in the Jinbei Mining Area and Shendong Mining Area gradually increases with the level of surface deformation, which is an inelastic deformation.

(2) The fluctuation in groundwater storage in the Loess Plateau decreased significantly, with a total reduction of 20.914 billion $m^3$ in 2020 and 2021. The groundwater storage in the Kubuqi Desert decreased by 359.42 mm. Drainage irrigation makes the seasonal variation in groundwater in the Hetao Irrigation District disappear, and irrigation encourages groundwater recharge. The total groundwater in the Hetao Irrigation District was reduced by 103.30 mm. The groundwater reserves in the northern Jinbei Mining Area are significantly recharged by precipitation, and the groundwater reserves were reduced by 45.60 mm. Precipitation and coal mining water consumption are the main factors for the reduction in groundwater reserves in Shendong Mining Area by 691.72 mm.

(3) The associations between surface deformation and groundwater storage in each landscape area are different. The temporal variation in groundwater storage and surface deformation are notably correlated when the micro-deformation occurs in the Kubuqi Desert, the Jinbei Mining Area, and the Shendong Mining Area. The cumulative correlation between surface deformation and groundwater storage in the Kubuqi Desert is 0.51. The surface deformation of the Kubuqi Desert is consistent with the fluctuation in seasonal groundwater reserves. Groundwater storage in the Hetao Irrigation District is affected by irrigation, and the increase in groundwater storage and soil water storage is the main factor in surface deformation. The response of surface deformation to the temporal change in groundwater storage in the Hetao Irrigation District lags by 1~2 months, with a cumulative correlation coefficient of 0.49. The cumulative change trend in mild surface deformation

and groundwater storage in the mining area of the Jinbei Mining Area is highly correlated (R = 0.57). The seasonal variation in groundwater reserves in Shendong Mining Area during severe deformation is significant, and the cumulative correlation coefficient is 0.67. Affected by the water-conducting fracture zone, the groundwater storage changes under severe deformation in the Jinbei Mining Area, and moderate deformation in the Shendong Mining Area has the shortest response time to seasonal precipitation and no lag period. The change in groundwater reserves and the fluctuation in surface micro-deformation in the mining area were significantly higher.

**Author Contributions:** Conceptualization, Z.L. and S.Z.; methodology, Z.L. and W.F.; software, S.Z.; validation, Z.L., L.H. and M.L.; formal analysis, Z.L. and X.Z.; investigation, Z.L. and S.W.; resources, Z.L., W.F. and L.H.; data curation, M.L.; writing—original draft preparation, Z.L. and L.Y.; writing—review and editing, Z.L. and S.Z.; visualization, M.L. and. S.W.; supervision, S.Z. and X.Z.; project administration, S.Z.; funding acquisition, S.Z. All authors have read and agreed to the published version of the manuscript.

**Funding:** Inner Mongolia Autonomous Region Science and Technology Achievements Transformation Project (2020CG0054) (Zhang, S.); Inner Mongolia Autonomous Region Science and Technology Plan Project (2020GG0076) (Huang, L.); the Basic Scientific Research Business Fee Project of Colleges and Universities Directly Under the Inner Mongolia Autonomous Region (JBYYWF2022001) (Zhang, X.); Inner Mongolia Natural Science Foundation (2021MS04013) (Zhang, X.); National Natural Science Foundation of China (52360032) (Zhang, X.); Young Faculty Research Capacity Enhancement Program (BR220118) (Zhang, S.); the Development Plan of Innovation Team of Colleges and Universities in Inner Mongolia Autonomous Region (NMGIRT2313) (Zhang, S.) and the Innovation Team of "Grassland Talents" (Zhang, S.) were jointly funded.

**Data Availability Statement:** The original contributions presented in the study are included in the article, further inquiries can be directed to the corresponding author.

**Acknowledgments:** We thank the anonymous reviewers for their constructive feedback.

**Conflicts of Interest:** The authors declare no conflicts of interest.

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
