# Peer review of "Associations between Surface Deformation and Groundwater Storage in Different Landscape Areas of the Loess Plateau, China"

_land, doi:10.3390/land13020184_

Round 1

Reviewer 1 Report

Comments and Suggestions for Authors

The article derived surface deformation over Loess plateau in China using SBAS-InSAR for ~two years (2020-2022) and studied it’s relation to the ground water changes derived from GRACE(+GLDAS,NLDAS) and in situ wells. The study explain how to the correlation varies among various land types (irrigated, desert and mining areas), extent of human influence and explored the nature of the correlation (elastic or inelastic). The study seems to a good fit for the journal. However, I have some important questions and suggestions that the authors need to address for publication.

Questions:

1.     Please justify the time period of the study. 2020-2021 seems rather a short period to properly bring out the correlation between ground water changes and observed surface deformation.

2.     In many areas, the observed surface deformation varies between positive and negative values meaning there is uplift and subsidence. While study mostly focusses on explaining subsidence, it ignores explaining the uplift. I recommend to include the reason for the uplift.

3.     Many times the term ‘invert’ is used in the context of InSAR and surface deformation. The authors need to clearly explain whether it is inversion of interferometric phase to timeseries of surface deformation or whether the surface deformation is inverted to infer any other geologic properties.

4.     The SBAS-InSAR method used is key for the entire study. In its current from in section 2.3.1, the method description lacks many significant details. I recommend, the authors to include details on the number of multilooks used, number of interferometric pairs used for each frame and average spatial coherence observed. A network diagram showing the interferometric connections can be included in the supplement.  

5.     The methodology doesn’t explain whether the authors have corrected for atmospheric noise and unwrapping errors in the InSAR processing. If surface deformation rates are computed from only 2 years of inteferograms, tropospheric noise can be very significant. I wonder whether the uplift observed is tropospheric noise.

6.     In its current form, the InSAR results presented in Fig. 3 looks to be suppressing noise. I recommend the color scale to be modified to bring various ranges of observed InSAR deformation. If the authors have used a specific software for interferogram generation and SBAS processing, please proved the details. The authors also need to specify, what are the coherence ranges considered for final results.

7.     The authors seems to have downsampled InSAR data to 0.01° grid and upsampled GRACE, FLDAS, GLDAS to the same. While downsampling of InSAR is understandable, upsampling data derived at 0.25° grid to 0.01° grid need to be explained more clearly on how it was done. I strongly recommend rewriting section 2.3.2 with clear explanations and include more references on studies from which the methods have been taken.

8.     In section 2.3.4, the authors say that they have used an ‘average specific yield’ value for each of the sub-areas. It may be valid, if the sub-areas have mostly unconfined aquifers. But if there are any confined aquifers and the draw-down from them is significant, such an averaging may not be valid. Please clarify this. Also I felt that terms like ‘average water yield’ or ‘average water supply’ were used interchangeably with ‘specific yield’. Please adjust this.

9.     In section 3.1.1, while comparing InSAR to levelling, it is not clear whether the authors used one InSAR pixel over the levelling location or averaged a group of pixels around the levelling location. It is mentioned ‘nearest neighbor pixel method’ in line 287. Does it mean the authors considered the nearest InSAR pixel to the levelling point location? If so, how near is that. I suggest the authors use mean of a group of InSAR pixels within a specified radius around the leveling point and compare. Also specify the radius value used.

10.  In section 3.3, again it is not clear what were the data-distance ranges used for comparison between surface distortion and ground water changes. If it was done on the 0.01 grid, please specify. If the correlation between ground water and distortion is lagged by 1-2 months in the irrigation area, can a correlation be given between ground water and time shifted surface distortion data. That might improve the R value for that area.

11.  In comparing the surface distortion to sub-surface processes in the mining areas, can a correlation be done with mining rates along with ground water changes?

12.  In section 4.2, lines 507-512, the infiltration rates also depend on soil permeability and water content.

13.     In the conclusion the authors say “Kubuqi Desert and Hetao Irrigation District belongs to elastic distortion”, does this mean the remaining areas (Jinbeai and Shendong) it is inelastic?

Suggestions:

1.     I suggest the authors consider changing “surface distortion” to “surface deformation” through out the study. Distortion can be man made changes to the landscape or erosion too.

2.     In Fig.1 and in some other areas, the authors refer to ‘Jinbei Mining area’ as ‘Shanxi mining area’. This is very confusing. Please change this.

3.     Please write the timeperiods with months as opposed to decimals. Ex: write as Jan. 2020 instead of 2020.1.

4.     Please increase the fontsize used in the figures. It is hard to read on a print version.

5.     Symbols and text overlap in Fig. 1c. Please avoid this.

6.     In section 2.3.1, the term ‘interference’ is used refer ‘interferogram’ or ‘interferometric phase’. Please correct this.

7.     Please include a figure showing the InSAR frames over the study area and show the InSAR reference point location/s in Fig. 3.

8.     In lines 315-316, the authors there are many ground fissures and land collapses in the study area. If so, it would be good to include some images of them.

9.     Please increase the font size in Fig. 7 or split it into multiple figures. May be study area wise.

Comments on the Quality of English Language

There are also some serious language mistakes which is making the manuscript hard to comprehend in its present form. I suggest the writing to be corrected for grammar and clarity before publication. I have included an annotated file with suggestions highlighted. Please go through them too.

Author Response

Thank you very much for your review suggestions and comments, we respond to your comments and suggestions one by one. Please find attached the review responses.

Reviewer 2 Report

Comments and Suggestions for Authors

Comments to the Author

RC: In general, the paper touches a topical subject that fits with the scope of the journal. The downscaling groundwater changes and surface distortion in the Loess Plateau from 2020 to 2021 were inverted, and the associations between the two under different human influences was analyzed according to different landscape at the same spatial and temporal scales. The scientific and methodological nature of the manuscript is vivid and has been fully cited. The draft is well written, and the parts are generally concise and rigorous, but the format (e.g., citation) needs to be standardized. In order to try to improve these problems, here are my opinions and suggestions. In my understanding, it is suitable for publication in Land after addressing the comments below.

Specific comments

1.      Line 34:

RC: Affected by 'water-preserved coal mining', … … . The reason should be written from the mechanism.

2.      Line 36:

RC: new technologies.  According to the main content of the article, re-refined.

3.      Line 52:

RC: Efficient and accurate detection of … … . The sentences are too long and difficult to understand.

4.      Line 122:

RC: The expression of the study area needs to be unified in full paper. (Region of Interest/ study area) 

5.      Line 141:

RC: Improve the graphic resolution of Fig. 1 and increase the font size in (b) and (c).

6.      Line 200:

RC: ta, tb variable should be italic

7.      Line 235:

RC: “where, … …”. Paragraph format needs to be modified.

8.      Line 380:

RC:  -46 -5 mm?  Expression unclear.

9.      Line 478:

RC: The expression of the graph in the text needs to be unified.

10.  Discussion:

Line 474: What is the elastic surface distortion? There seems to be no introduction here, only the last sentence appeared the term, but the role of elastic surface distortion and inelastic surface distortion should be equally important.

11.  Line 558:

RC: There are significant ……. This sentence can be further improved to facilitate readers ' reading.

12.  Conclusion:

RC: Revise the conclusion to make it clearer

Author Response

(The authors gave the same response as above.)

Reviewer 3 Report

Comments and Suggestions for Authors

The study investigates the associations between surface distortion and groundwater storage changes in different landscape types on the Loess Plateau in China, specifically focusing on the Kubuqi Desert, Hetao Irrigation District, Jinbei Mining Area, and Shendong Mining Area. Utilizing Sentinel-1 and GRACE data from 2020 to 2021, the research explores the rates of surface distortion and groundwater storage changes, emphasizing the need for understanding these dynamics in the context of regional environmental safety and water resource management.

The study addresses a critical issue in the Loess Plateau, where surface distortion impacts groundwater storage. The use of Sentinel-1 and GRACE data adds a novel aspect to the research, enhancing the understanding of this dynamic relationship. Considering these aspects, the study holds merit for publication, offering valuable insights for researchers and policymakers concerned with regional environmental dynamics and water resource management.

However, the findings of the study have limited generalizability. The study is specific to the Loess Plateau, and while this specificity is essential for regional considerations, it may limit the generalizability of findings to other geographical locations.

While Sentinel-1 and GRACE data provide useful information, the study might benefit from a discussion on the resolution limitations, and how the different resolutions may have contributed to potential uncertainties of the outcome.

How was the integration of Sentinel-1 and GRACE data performed to derive surface distortion and groundwater storage changes? Additionally, what measures were taken to ensure the accuracy of the integrated data, especially considering the different sources and resolutions?

The study mentions the relevance for environmental safety and water resource management but lacks a detailed discussion of the practical implications or policy recommendations based on the findings.

The surface distortion verification results indicate variations between measured and inverted distortions. Can you provide insights into the factors contributing to the observed errors and how these might impact the overall accuracy of the surface distortion measurements?

The study establishes a correlation between groundwater storage changes and surface distortion. How robust is this correlation, and are there specific instances where the relationship is more pronounced or less evident? Additionally, how does the correlation contribute to understanding the interactions between surface and subsurface processes?

The discussion lacks a section addressing the limitations of the study. Acknowledging potential constraints, uncertainties, or data gaps would enhance the credibility of the findings and provide a more balanced perspective.

Ensure consistent usage of terminology throughout the discussion. For example, the term "distortion grade" is used but not defined. Providing clear definitions or explanations for specialized terms will help readers, especially those outside the immediate field.

The discussion could conclude with a synthesis of the key findings, emphasizing their significance and potential implications. This would help readers grasp the main takeaways from the study.

the figures have low resolutions. Please modify. 

Additionally, I have some minor issues in the abstract and introduction sections. 

  1. 1-The abstract appears lengthy. I recommend using a more concise language to present only the key findings.

  2. 2- In the introduction, particularly when introducing remote sensing, consider using a more general language. Also, please review the text for any language-related issues.

Comments on the Quality of English Language

English is good, but some minor issues were detected. 

Author Response

(The authors gave the same response as above.)

Reviewer 4 Report

Comments and Suggestions for Authors

The paper discussing the associations between surface distortion and groundwater storage in different landscape areas of the Loess Plateau, Chinaseems technically sound. However, a few potential areas might require clarification and improvement.

Line 14: Typo - "mansucprit" instead of "manuscript."

Line 24: Inconsistent spacing in numeric ranges, e.g., "-5 - 5 mm/yr."

Line 26: Inconsistent negative signs, e.g., "−95.33 - 26 mm/y."

Line 28: Typo - "Kubuqi Desert decreased by 359.42 mm."

Line 32: Unclear sentence - "this study inverted and verified the sur- 19 face distortion."

Line 34: Typos and missing spaces, e.g., "Hetao irrigation area."

Line 36: Punctuation error, unclear sentence structure.

Line 44: Typo - "chiefly" instead of "chief."

Line 46: Sentence structure issue - "disorderly open up of resources" needs clarification or correction.

Line 161: "National Meteorological Science Data Sharing Service Platform-China Surface Climate Data Daily Dataset" - The text might contain some inconsistencies or be a bit lengthy. Consider revising for better readability and clarity.

Line 165: "from 2020.1 to 2021.12 months" - It might be more appropriate to state "from January 2020 to December 2021" for clarity.

Line 172: "2020.3.27" could be rephrased as "March 27, 2020."

Line 180: "greater than 12 months" could be changed to "exceeding 12 months."

Line 182: "adjacently" could be rephrased as "adjacent."

Line 285: Replace "2021.1.21" with "2021.1.25" for consistency in date format.

Line 292: Add a space after "12.9" in "12.9 mm" for proper formatting.

Line 295: Change "2021.12.4" to "2020.12.4" for consistency in the year.

Line 298: Add a space after "Fig.2g," to separate the reference from the text for clarity.

Line 322: Adjust "Halagou mining area" to "Halagou Mining Area" for consistency.

Line 368: Adjust "spacial" to "spatial" for correct spelling.

Line 385: Change "Kubuqi" to "Kubuqi" for accuracy.

Line 388: Adjust "surface time series distortion" to "surface distortion time series" for clarity.

Line 390: Add "shows" after "comparative analysis" for better syntax.

Line 390: Consider rephrasing "cumulative reduction of groundwater reserves in 2020 is the main" for clarity.

Line 392: Revise "temporal distortion of groundwater reserves and surface distortion" for clearer expression.

Line 394: Replace "from May to October 2020 are more significant" with "between May and October 2020 is more pronounced" for better clarity.

Line 396: Revise "subsidence was significant from March to April" to "exhibited significant subsidence from March to April" for clearer phrasing.

Line 398: Consider rephrasing "through the analysis of cumulative distortion" for better clarity.

Line 402: Adjust "micro-time series distortion" to "micro-time series of distortion" for clearer expression.

Line 408: Revise "consistency between the change of groundwater reserves and surface distortion" to "consistency between changes in groundwater reserves and surface distortion" for clarity.

Line 410: Adjust "fluctuated and settled" to "fluctuated and settled down" for clearer phrasing.

Line 412: Revise "trends was significant" to "trends was notable" for better expression.

Line 449: Consider revising "stor-" to "storage" for completeness.

Line 454: Adjust "wetting effect to accelerate the surface subsidence of the gully slope with" to "wetting effect, accelerating the surface subsidence of the gully slope with" for clearer phrasing.

Line 456: Consider revising "surface uplift" to "surface elevation" for better expression.

Line 458: Adjust "time series distortion is consistent" to "time series distortion are consistent" for subject-verb agreement.

Line 461: Adjust "United States, the depletion" to "United States, has led to the depletion" for clearer context.

Line 463: Revise "is elastically deformed" to "undergoes elastic deformation" for clearer expression.

Line 465: Consider rephrasing "stress changes geological structure" for clarity.

Line 470: Adjust "time series distortion" to "time series distortions" for plural consistency.

Line 472: Consider rephrasing "water storage through 'water-preserved coal mining'" for better clarity.

Line 474: Adjust "consistent with fluctuation" to "consistent with fluctuations" for clarity.

Line 476: Consider revising "first half of the year [57]" for better contextual placement.

Line 478: Adjust "found that after the coal mine was closed" to "noted that following the closure of the coal mine" for better clarity.

Line 480: Revise "uplifted or subsided under" to "experienced uplift or subsidence due to" for better expression.

Author Response

(The authors gave the same response as above.)

Round 2

Reviewer 1 Report

Comments and Suggestions for Authors

I feel the authors have reasonably addressed my comments. The language quality seems to have been improved. I appreciate the effort of the authors.